# The Violent Aspect of Widowhood Rites in the South African Context

**Ratidzai Shoko * and Sizakele Danke**

Educational Foundations, University of South Africa, Pretoria 0003, South Africa; danksm@unisa.ac.za
* Correspondence: shokor@unisa.ac.za

**Abstract:** In African culture, widowhood is frequently accompanied by rites that must be carried out by the widow. Widows are compelled to carry out these rites and may not feel comfortable executing them since they involve violence. The minority who dares to refuse to participate can face serious consequences because they are persecuted by their families and society. Research shows that widows suffer from fear and coercion, stigmatisation, dehumanising experiences, movement and social restrictions, and exposure to harmful traditional practices. This article examines violent aspects of widowhood rites within the South African context. A qualitative study that examined oppressive structures and how they impacted social injustice and the marginalisation of widows was employed. The data were collected from a purposeful sample of widows in Gauteng province, South Africa. Semi-structured interviews were used to gather data from 28 widows, which were then subjected to thematic analysis. Our findings showed that widows were subjected to painful widowhood rites, which were frequently performed against their preferences. The rites affected them both physically and emotionally. The article recommends that policies be put in place to safeguard the rights of widows and protect them from exploitative cultural beliefs.

**Keywords:** African; culture; tradition; violence; widowhood; widowhood rites

## 1. Introduction

### 1.1. Widowhood Rites in Africa

Widowhood is a stage in a person's life where one loses a spouse by death and has not remarried (Brown 2016). The term "widowhood" describes the status of having lost one's spouse to death. A widow must overcome several obstacles during this stage of life. Widows may face various forms of discrimination that are often rooted in cultural and social norms. These widows experience social isolation where the widows are stigmatised within their communities and excluded from social activities. The widows may also be exposed to harmful traditional practices such as injurious rituals, restrictions, or expectations. They may also experience multiple forms of discrimination based on a combination of factors such as gender, age, socioeconomic status, and cultural background (Saif-Ur-Rahman et al. 2021). The death of a spouse leaves the widow vulnerable in various ways.

Widowhood rites are performed on the widow as part of the widowhood process throughout sub-Saharan Africa and these are detrimental to the widow (Ude and Njoku 2017). According to Brown (2016), the basis of various widowhood rites, which aim to end the marital ties between the dead and the living, is the strong belief that the spirit of the deceased still views the widow as his wife and may wish to carry out his marital and conjugal duties with her. However, some rites may be forced on the widow (Amoo et al. 2022), and others may be physically unsuitable and endanger the widow's life (Iloka 2022) as they include a violent element. When violence is used, it takes place without respect, sympathy, or consideration for the widow in her time of loss. Conversely, Kenny (2020) states that most people empathise with widows and believe that they deserve sympathy. Although some

people may sympathise with the widows, there is a need for them to advocate for changes in the way that society treats widows and in the application of widowhood rites.

According to a 2015 World Bank report, there are an estimated 258 million widows in the world. One in ten African women over the age of 14 is widowed (World Bank 2018). 10.2% of South Africa's entire population is widowed, according to data acquired in 2022 (Statista 2023). Despite the high number of widows, they frequently face marginalisation and trauma (Banford Witting et al. 2020). In most of Africa, marriage serves as the only legal foundation for a woman's access to social and economic rights, which are lost following divorce or widowhood (World Bank 2018). The high percentage of widows suggests that much work needs to be undertaken to raise their status and protect their interests. The ill-treatment of widows adds to social injustices and stigmatisation. Gender equality is the focus of Sustainable Development Goal 5 (SDG 5), which seeks to eliminate all forms of violence and harmful behaviours against women and girls (United Nations 2015). Rituals associated with widowhood are among the customs that may be damaging to women (Nwadialor and Agunwa 2021). As a result, it is debatable whether the 2030 deadline for achieving gender equality and women's empowerment will be realised. There is still a long way to go before widow empowerment can be accomplished if widows are unable to voice their concerns about how these widowhood traditions affect their way of life.

Widowhood in the Southern African region is embedded in sociocultural contexts that shape the experiences and roles of widows (Cebekhulu 2016). While it is important to note that the Southern African region is culturally diverse, some common sociocultural changes and challenges related to widowhood can be identified. The sociocultural changes include changing gender roles, legal reforms, non-governmental organisations (NGO) and activist interventions, health and well-being, and the educational opportunities that widows have. Even though the experiences may vary, it can be argued that all African widows, whether they are educated or regardless of their social standing, experience some type of violence. There is still a dearth of scholarly investigation into widows, notably in South Africa. The widowhood rites in South Africa have received minimal attention in the literature, as little is known about their violent characteristics. In the South African context, widowhood rites may also be influenced by the specific cultural practices of different ethnic groups within the country. The experiences of widows and the associated rituals can differ among various South African communities.

Understanding the specifics of widowhood and widowhood rites in South Africa would require consideration of the cultural and traditional practices within a particular community or ethnic group. It is essential to approach these topics with sensitivity, recognising the diversity and uniqueness of South Africa's cultural landscape. Research conducted by Khosa-Nkatini et al. (2020) among the Tsonga showed that one of their widowhood rites includes being cut on the private parts using a razor so that when the widow bleeds, the husband's blood is thus removed from her body. Khosa-Nkatini et al. (2020) furthermore argue that these rituals are oppressive to women. Widowhood rites have been discussed in the current literature; however, this article focuses on their violent elements in the context of South Africa. The article made use of data gathered from widows in Gauteng province. To comprehend the violent elements of these rites, these data were compared with the extant literature on widowhood rites.

This paper aims to highlight the violent aspects of the widowhood rites. The paper will also discuss what can be done to enhance the welfare of widows in South Africa and protect them from harmful cultural practices. This article will explore the literature related to the study, the theoretical framework, and the findings, conclusions, and recommendations of the study.

Widowhood is a social status associated with some institutionalised cultural taboos, social conventions, and attitudes toward women who have lost their spouses or partners (Magudu and Mohlakoana-Motopi 2013). There is a notion that the widow must carry out certain rites following the death of her husband. Widowhood rites are practices and ceremonies that have been practiced for a long time in various cultures to signify the

change in a person's status from married to widow. Widowhood rites in Africa have deep historical and cultural roots, and they vary significantly across the continent due to the diversity of ethnic groups and traditions. These rites are often embedded in cultural beliefs, social norms, and religious practices. The experience of widowhood is influenced by cultural and societal factors, as evidenced by studies exploring the lived experiences of widows in different contexts such as rural Nigeria and South Africa (Ugwu et al. 2020; Motsoeneng 2022). While there is diversity, some common themes and practices can be identified in various regions of Africa.

In African culture, being a widow frequently evokes two feelings, shame, and pity, based on the woman's location, social connections, age, status, class, or ethnicity (Fasanmi and Ayivor 2019). The widow is ashamed of her new marital status and receives pity from society. Many African cultures have specific mourning periods during which widows are expected to grieve for their deceased spouses (Bauta 2022). The duration of these periods can vary. Furthermore, widows may be required to undergo symbolic changes in appearance, such as shaving their heads, wearing specific clothing, or using particular accessories. In some societies, widows may be required to live in isolation or seclusion during the mourning period to allow the widow time to grieve and also to protect the community from any perceived negative spiritual effects associated with death (Adeyemo 2016). In some cultures, widows may be expected to go through specific rituals before they can remarry. These rituals may be seen as a form of cleansing or purification. The level of community support for widows can vary. While some societies provide emotional and practical support, others may stigmatise widows, viewing them as a source of bad luck or associating them with negative spiritual forces.

In African culture, rituals are a very delicate subject because they represent an integral element of an individual's identity and sense of pride in being African (Khosa-Nkatini et al. 2020). The widows feel that cultural customs are to be observed to please their ancestors, even if they are aware that their rights are violated (Mabunda and Ross 2023). The grieving women are required to participate in the beliefs because they are seen as societal rites (Brown 2016). Therefore, the widows are expected to do what society views as proper without question. Brown (2016) further suggests that African women are required to complete these rites successfully, regardless of how gruesome or humiliating they may appear to an outsider. This is considered courage, a sign of love for the departed husband, and a way of absolving the widow from any responsibility for his death. As a result, widows comply with numerous rituals primarily out of responsibility and not because they find them to be amusing (Manala 2015). It removes the widows' autonomy because they are forced to conduct these rites.

When women lose their spouses, they are expected to show that they are grieving. According to Brown (2016), the expression of grief can take on many different forms, such as prolonged sobbing and wailing, donning the appropriate mourning apparel, forgoing meals, personal hygiene, grooming, shaving one's hair, and walking barefoot, among other things. Even while widowhood rites differ from community to community, Shahin (2022) noticed that there is a similar theme running through them all: the infliction of physical, mental, and psychological harm on widows. Inflicting harm on an individual constitutes violence.

Strong social rites, practices, and taboos associated with widowhood exist in many South African social and cultural groups and are influenced by powerful patriarchal systems and religious beliefs in society (Magudu and Mohlakoana-Motopi 2013). According to Khosa-Nkatini et al. (2020), women are socialised to be submissive and quiet in all areas of their marriage and womanhood. This kind of socialisation is related to how widows react if detrimental ceremonies are performed. The different cultural groups have common ways of conducting these rites, with a few deviations. According to research by Mabunda and Ross (2023), South African widows are subjected to restrictions on their freedom of movement, participation in specific activities, modes of dress, sitting arrangements, the use of distinct utensils, and cleansing rituals. According to a study conducted by Khosa-Nkatini et al. (2020) on Tsonga widows in South Africa, widows carry out the rites out of fear of

what might happen if they do not. Excessive fear may cause psychosomatic problems for the widows.

Other African widows may be subject to widowhood rites, which are not exclusive to South Africa. In Nigeria, this includes drinking the water used to wash the corpse, sleeping in the same room where the corpse is laid, being confined to a room and made to sit on ashes, being served food on broken plates, and, in some cases, being forbidden from looking at the person who served the meal, as well as being limited to wearing specific colours, styles, or tattered clothes for some time (this varies across cultures) (Ajayi et al. 2019). Depending on the tribe and region where these rites are performed, the rites take on many different forms. Widows in Nigeria's Igbo culture are barred from touching anything, even their bodies, and are instead given sticks to scratch themselves (Brown 2016). Additionally, according to Brown (2016), after the cleansing ceremonies are finished, everything they touch during the grieving is burned. These rituals imply that the widow must be purified since she is impure. The widow is brought to the stream, given a ceremonial wash, has her hair cut short, and all her mourning-related attire is either burned or tossed into the forest (Brown 2016). The rites are similar to the rites conducted on South African widows and other widows in sub-Saharan Africa. Widowhood rites in Africa exhibit both similarities and differences across various cultures due to the continent's rich diversity of ethnic groups, traditions, and belief systems. While there are common themes, the diversity of African cultures means that practices can vary widely from one region to another. Similarities include the mourning periods, symbolic changes, rituals and ceremonies, and community involvement. Although there are similarities, differences may exist in the specific ritual practices, isolation practices, property and inheritance rights, gender dynamics, community attitudes and support, and economic implications.

According to Brown (2016), it is impossible to ignore the fact that these rites are inherently harmful to the widow's physical and mental health. Conversely, Tasie (2013) asserts that in the African setting, widowhood ceremonies are largely intended to safeguard the widow and not to dehumanise or harm her; they are essentially intended for the widow's overall well-being. The claims support Okoro's (2018) assertion that widowhood rituals and rites are not meant to dehumanise any person. According to Odimegwu (2000), traditional African widowhood practices are connected to ideas about death, gender roles, inheritance, family ties, and family structure. Widows are therefore required to perform these rites since societal beliefs place a high value on them. According to Brown (2016), widows consent to the mourning rites and practices because failure to do so would be seen as disrespectful to their late spouses, to themselves, and the community. Since these ceremonies are carried out immediately after a husband's passing, widows are likely to be in distress; therefore, carrying them out would be the greatest way to honour his memory. They are compelled to follow these rites and practices without question, no matter how painful and upsetting they are for them because they are seen as transmitters and carriers of bad luck and contamination, who are harmful to the community (Brown 2016). The widows are forced to cooperate as they have no other option.

### 1.2. Violent Forms of Widowhood Rites

Violence within widowhood rites refers to any physical, emotional, or psychological harm inflicted upon a widow as part of the cultural or traditional practices associated with the transition from marriage to widowhood. While many widowhood rites are intended to be symbolic or cultural expressions of mourning and transition, some practices can be harmful and violate the human rights and dignity of widows (Nwadialor and Agunwa 2021). It is crucial to recognise that not all widowhood rites involve violence, and many communities have evolved their practices to be more supportive and respectful of widows. However, harmful practices persist in some ethnic groups, and efforts to raise awareness, challenge harmful traditions, and promote more humane and dignified approaches to widowhood need to be implemented.

The South African Constitution (Republic of South Africa 1996) and the Convention on the Elimination of All Forms of Discrimination Against Women (CEDAW) both prohibit discrimination against anyone based on gender (Magudu and Mohlakoana-Motopi 2013). These policies safeguard everyone, including widows. According to Dowuona-Hammond et al. (2020), widowhood rites constitute a disability. This is due to how incapacitating they are. Gender-based violence, which includes gender discrimination, domestic abuse, sexual assault, conflict-related violence, and battlefield rape, is experienced both personally and systemically (Yadav and Horn 2021). Violence happens in a variety of settings, and this article focuses on the violence that takes place when widowhood ceremonies are performed. Understanding how people's lived experiences are simultaneously personal and political depends on how we conceptualise violence (Yadav and Horn 2021). With this perspective, it is probable that some people, if they do not engage in reflecting on lived experiences, will not notice anything wrong with the rites that widows must face. The fact that these violations still occur in South Africa shows that the problem is not necessarily a lack of political will or appropriate legislative or policy frameworks, but rather that the current legal provisions are not being applied and enforced effectively and efficiently (Magudu and Mohlakoana-Motopi 2013). According to the World Bank (2018), giving widows and divorcees a secure place in society is essential to the greater fight for gender equality. The widows would obtain their rights if the community in which they reside began by upholding the applicable legal provisions.

There are various reasons for performing widowhood ceremonies, and each has an impact on the widows. Many widows carry out widowhood ceremonies because they worry about the potential consequences if they do not (Brown 2016). These traditions frequently severely restrict women's access to privacy when they are mourning the loss of their spouses, husbands, or lovers (Magudu and Mohlakoana-Motopi 2013). These rituals could be performed to win the approval of family ancestors and the respect of the community (Baloyi 2017). Many widowhood traditions, despite their seeming cultural appropriateness, are harmful to the widows' health. If personal hygiene is neglected, one is more likely to contract diseases such as skin rashes, vaginal infections, and urethra infections. They could be more susceptible to respiratory infections and colds if they are dressed improperly. Some of the ceremonies that are carried out at night could put the widows in danger of attacks from animals or criminals. As a result, widows are unable to meet many of their basic physical needs, including those for food, housing, and sleep. According to some of the research's findings, women were handed eggs to break with their thighs which required using force and which caused pain. The widow must also jump over a fire as part of a rite. If she is burned by the fire, it indicates that the ancestors are upset with her for killing her spouse (Khosa-Nkatini et al. 2020). However, the widow might not jump because she is unable to jump or because she is too worn out from sorrow. The widow is put at risk by this rite and is made to feel fearful. For the Tsonga people, ceremonies must be performed after death for the ancestors to accept a new member into the ancestral tribe; otherwise, the deceased's ghost will linger around the home and there will be no peace in the family (Khosa-Nkatini et al. 2020). Another widowhood ritual involves the widow putting out a fire using her urine. In addition, elderly women would then slash the vulva of the widow until she bled in the belief that this would rid her of the husband's blood as part of the Tsonga widowhood rituals (Khosa-Nkatini et al. 2020).

The person in mourning requires compassion, love, understanding, support, and counseling (Brown 2016). The widows are, however, immediately ushered into strict widowhood customs, which exacerbate the grief process. Widows' experiences of extreme anxiety constitute severe psychological trauma (Brown 2016). The widows are continually reminded through rituals that their dead husbands' souls are after them and want to have sex with them or hurt them (Brown 2016). The idea might make people avoid widows, which would exacerbate loneliness and depressive feelings. Their inability to adjust to life without their spouses, particularly with respect to how to raise their children, may be one of their psychological difficulties. Anorexia, lack of sleep, and stress may result from this.

The primary goal of the social concept of wife inheritance is to prevent the widow from going without, but often, greed on the part of the deceased person's relatives trumps this noble goal, as they are more concerned with accessing the estate of the deceased person than they are with taking care of the widow and her children (Brown 2016). Looting may precede this, which disregards the widow's combined or independent possessions. According to Effeah et al. (1995), widowhood is the ideal opportunity to settle old grudges, particularly in cases where the widow was never treated favourably by her late husband's family. Thinking of the widow as a piece of property that must be inherited is degrading.

Gender intersects with various social relations of power to shape the nature and impact of widowhood rites in complex ways. Widowhood rites are deeply embedded in cultural, social, and economic contexts, and they often reinforce or challenge existing power dynamics (Makanga 2022). Many societies in Southern Africa are patriarchal, where men hold primary positions of power. Widowhood rites, often rooted in tradition, may reinforce existing gender inequalities by subjecting widows to practices that reflect and perpetuate patriarchal norms (Uwadineke and Umunna 2022). Widows may be subjected to rituals influenced by cultural and religious beliefs that reinforce traditional gender roles. These beliefs often dictate how widows should behave, dress, and participate in community life, contributing to the maintenance of gender norms. Widowhood rites can contribute to social stigma, and the perception of widows may be influenced by prevailing gender norms (Motsoeneng and Modise 2020). Social expectations regarding widow behaviour, dress, and remarriage can be particularly gendered, reinforcing stereotypical views of appropriate female conduct.

The experiences of widows are also shaped by intersecting social identities, such as age, ethnicity, and socioeconomic status. For example, a young widow may face different expectations and challenges compared to an older widow. Intersectionality highlights the interconnectedness of various power dynamics in shaping individual experiences (Collins 2019). Understanding how gender intersects with other social relations of power in the context of widowhood rituals is crucial for developing interventions that promote gender equality and address the specific challenges faced by widows in diverse cultural settings. It requires a holistic approach that considers the interplay of cultural, economic, legal, and social factors.

While discussions often focus on the challenges and inequalities faced by women, it is important to recognise that women can also enjoy privileges based on various factors, including gender, widowhood, community, and human status/identity in the contemporary world. It is crucial to note that the extent and nature of these privileges can vary widely depending on cultural, social, economic, and geographic factors.

In certain cultures, widowhood can bring about a shift in social status for women. Some communities may empower widows, providing them with support and resources to live independently. This empowerment can lead to increased decision-making power and autonomy for widowed women.

Women may benefit from strong community networks that provide social support, mentorship, and opportunities for collaboration. These networks can enhance their personal and professional development and contribute to their overall well-being. In some cases, women may be the primary beneficiaries of humanitarian and development initiatives aimed at empowering vulnerable groups. This could include access to healthcare, education, and economic opportunities.

Some societies may uphold positive gender norms that recognise and appreciate the contributions of women in various spheres. This can lead to increased respect and recognition for women's roles and achievements. The global women's rights movement has contributed to increased awareness and advocacy for gender equality. Women may benefit from the progress made in terms of challenging gender stereotypes and promoting equal opportunities.

It is important to acknowledge that these potential privileges are not universal, and many women continue to face significant challenges and discrimination. Additionally, the

intersectionality of identities, such as race, ethnicity, socioeconomic status, and more, can further influence how women experience privilege or disadvantage. Understanding the complexities of these issues is essential for promoting a more inclusive and equitable world for everyone.

### 1.3. Intersectionality: A Theoretical Framework

Intersectionality is a concept developed by legal scholar and critical race theorist, Kimberlé Crenshaw, in the late 1980s. It highlights the interconnected nature of social categories such as race, gender, class, sexuality, and other forms of identity, recognising that individuals may experience overlapping and intersecting forms of oppression or privilege (Cerezo et al. 2020). According to Atewologun (2018), intersectionality is a crucial framework that gives us the perspective and vocabulary to examine the links and interdependencies across social categories and systems. Intersectionality examines the connections between oppressive structures that prevent their independent analysis. The intersectionality theory, according to Kuran et al. (2020), focuses primarily on how the exercise of power, through intersecting dominance and oppression, impacts people who experience various social injustices, with the resultant multiple marginalisation. Intersectionality brings attention to the experiences of marginalised and often overlooked groups, in this case, widows. It encourages a more comprehensive understanding of social issues by considering the perspectives of those at the intersections of multiple identities (Avraamidou 2020). Widows' rights are human rights, just as women's rights are. Women cannot be said to have rights while some of them are still subject to oppression, which goes against this idea. The theory explains why many forms of inequality exist on their own and how they interact to produce new kinds of oppression. Intersectionality emphasises that individuals have multiple social identities, and these identities are interconnected (Nair and Vollhardt 2019). It also acknowledges that systems of power and oppression are interconnected and mutually reinforcing. It recognises that individuals may experience various forms of discrimination and privilege simultaneously. It acknowledges that each individual's experience is unique and complex, and it is shaped by the intersections of their various social identities.

According to Kelly et al. (2021), intersectionality suggests that identities such as gender, race, sexual orientation, and other markers of difference intersect and reflect significant societal structures of oppression and privilege including sexism, racism, and heteronormativity. Intersectionality focuses on structural inequalities embedded in societal systems. It goes beyond individual experiences to examine how social, economic, and political structures contribute to and perpetuate oppression or privilege. It can help us comprehend how different oppressions are felt by different oppressed persons depending on their intersecting identities. Theoretical explanations of how heterogeneous members of groups (such as women) may experience the workplace differently depending on their ethnicity, sexual orientation, class, and/or other social locations are provided by intersectionality, which advances analytical sophistication (Atewologun 2018). Within any group, there are differences in experiences based on intersecting identities. Intersectionality recognises that individuals within a particular social group may have different levels of privilege or face varying forms of discrimination based on additional factors (Holman and Walker 2021). Multiple discrimination emphasises the idea that the combination of various social identities can create unique and compounded forms of disadvantage. Intersectionality recognises that individuals are shaped by a combination of social factors, and the effects of discrimination or privilege are often amplified when multiple aspects of identity intersect (Burke et al. 2023). People experiencing multiple discrimination may face compounded disadvantages, as each layer of their identity can contribute to a unique set of challenges and obstacles. Not all individuals within a particular group or with a shared identity have the same experiences. Multiple discrimination emphasises the diversity of experiences within marginalised groups, considering the various intersections of identity.

While women battle for their rights, widows may encounter oppression in a new way since they, in addition, suffer from being widowed. According to Carastathis (2016), inter-

sectionality contends that all forms of marginalisation are significant and that oppression and privilege can change depending on the situation. Different social groupings are not independent but rather impact one another due to their interdependence and interaction. Intersectionality serves as a framework for social justice advocacy. By understanding the intersecting nature of oppression, advocates can develop more inclusive and effective strategies to address systemic inequalities. By increasing awareness of concerns of social fairness and inequality in companies and other institutions, sensitivity to such discrepancies increases the likelihood of social change (Atewologun 2018). As a result, it can demonstrate how gender may affect how widows are handled. A person cannot be comprehended or viewed in isolation from their social group identities. Discrimination can take on various forms depending on the intersections. Widowed females may face discrimination. Intersectional sociology examines the social challenges and difficulties that different groups face both independently and concerning one another. The intersectionality theory is used in this article to present a clearer and more complex image of vulnerabilities and vulnerable groups (Kuran et al. 2020); it serves as a guiding concept in risk and crisis management. The theory aids in understanding how social interdependencies support widow oppression and how they obstruct independent examination of how cultural practices might be taking advantage of widows. The intersectional perspective reveals how power is exercised to coerce widows into participating in these widowhood practices. The nature of widows' vulnerabilities must be understood because they are a vulnerable population. It is critical to comprehend the type of harsh customs they may encounter with widowhood traditions. This article seeks to examine the violent aspects of the widowhood rites.

## 2. Methodology

The phenomenological case study that was used in this article is based on the idea that reality is socially created by how different people or groups define a given circumstance (Berger and Luckmann 2023). The interpretive paradigm was employed, and it aims to comprehend the underlying meanings and patterns while offering a thorough and vivid depiction of lived experience. It places a strong emphasis on comprehending the individualised meanings and interpretations that people give their experiences. Through the gathering of qualitative data, the experiences and meanings associated with widowhood rites are examined on an individual and societal level in this paper. The data used for this article were part of a broader research project on "Widowhood issues in South Africa (WISA) conducted in marginalised communities in Gauteng province". Individual semi-structured interviews were used to gather the data. The interview questions were mainly related to their demographic information and the experiences of widowhood from the time the husband passed on. The interviews were 45 min to one hour in length and were conducted by members of the WISA research team. The interviews were conducted by six female members of the research team. Because widowhood is such a delicate topic, the widows felt uncomfortable being interviewed by the research team's male members. The positionality of the research team influenced the data collection and analysis. Three members of the research team were widowed and shared certain commonalities with the participants; therefore, they were able to build a more honest and trustworthy relationship. Because the research team has experience with qualitative research, positionality influenced the choice of data collection techniques, which is why this study was qualitative. There were two psychologists in the research team to handle issues because widowhood is a delicate topic and there was a likelihood that widows would weep when they remembered the traumatic experiences. The interview participants gave their permission for the recordings to be made. The data collected were analysed thematically. Researchers engaged themselves in the data before finding themes. To fully comprehend the subject matter, this required reading transcripts, notes, or other sources of qualitative data several times. Researchers used open coding during the first round of coding, assigning labels or codes to various data segments. Next, categories and subcategories were created using the codes. As a result, these codes were grouped into more general themes that highlighted recurring trends in

the data. To guarantee the validity of the findings, member verification was conducted. The rigor and trustworthiness of the results were improved by the researchers working together during the analytic process. Because the coding was conducted as a team, there was a chance that there might be a coding mismatch during the process. The researchers followed a precise coding procedure and coded at a single location, which allowed for discussion and consensus-building when it came to resolving these differences. The study population comprised 28 widows in marginalised communities in the Gauteng province of South Africa. The widowhood status was the only criterion used and these widows were widowed within the last ten years (2013 to 2023). The table below (Table 1) shows the demographic information of the participants.

**Table 1.** Participants' demographic information.

| Pseudonym | Age | Ethnic Group |
|---|---|---|
| Tebby | 68 | Venda |
| Mye | 62 | Tswana |
| Mary | 66 | Tsonga |
| Selly | 53 | Tswana |
| Wish | 50 | Pedi |
| Mbali | 45 | Tswana |
| Pretty | 65 | Pedi |
| Juliana | 62 | Xhosa |
| Blessing | 54 | Venda |
| Sonto | 40 | Xhosa |
| Veli | 52 | Xhosa |
| Musa | 44 | Zulu |
| Thabo | 54 | Zulu |
| Dumi | 49 | Tsonga |
| Thato | 46 | Zulu |
| Zanele | 60 | Tsonga |
| Tilda | 53 | Tsonga |
| Bongi | 49 | Zulu |
| Prim | 51 | Xhosa |
| Kara | 35 | Tswana |
| Kea | 47 | Sotho |
| Tshepo | 55 | Pedi |
| Tshego | 61 | Sotho |
| Lebo | 43 | Tswana |
| Tyra | 42 | Pedi |
| Kamo | 51 | Tswana |
| Nta | 52 | Tswana |
| Kago | 48 | Pedi |

To find and choose the widows, the snowballing method was applied. In other words, someone who meets the requirements to be included in the study recommended additional people who also meet the requirements. This method was appropriate because the information was gathered about widowhood; therefore the participants should have

been widows who had participated in widowhood rites. The information was obtained using a study sample from the population because it was impractical to include all widows in Gauteng. The widows were from different ethnic backgrounds which gave the researcher an insight into how different ethnic groups conduct their widowhood rites. Ethical clearance to conduct research was obtained from the university's ethics committee under whose auspices the research was conducted. Before conducting the interviews, the researcher obtained written consent from the participants. The participants were informed about the research purpose and procedures and that they were free to refuse to answer any questions or to withdraw anytime during the research without penalty. Pseudonyms were used to maintain the anonymity of the research participants.

## 3. Findings

Numerous widowhood traditions are imposed on women whose husbands have passed away, making them vulnerable. These rites differ across different cultural and ethnic groups. These rites include drinking the water used to wash the corpse, sleeping in the same room where the corpse is laid, being confined to a room and forced to sit on ashes, being served food on broken plates, and, in some cases, being forbidden from looking at the person who served the meal; furthermore, in some cases (this varies across cultures), widows are limited to wearing certain colours, styles, or torn clothing for some time (Ajayi et al. 2019). In addition, these widows may occasionally be expected to scream to be heard by others in the neighbourhood (Ajayi et al. 2019). This also serves to show that the widow is mourning her husband.

The following themes emerged from the study: instilling fear and coercion; exposure to harmful conditions; dehumanising experiences; stigmatisation; and amplification of injustice.

### 3.1. Instilling Fear and Coercion

It was discovered that widows are forced to participate in widowhood rites out of fear. Tebby stated:

> "My hair was cut, and I was instructed to place eggs between my legs; if the eggs fell, it meant that I had been having intercourse immediately after my husband's passing before the cleansing was completed".

She further elaborated:

> "I am Venda, and I was married to a Tsonga. I had long hair which I had kept for a long period, and I was not happy that it was cut. I was told that if I do not do those rituals the husband will come and sleep with me".

There was no consensus in conducting these rites. The widow can be accused of something they did not do if an egg falls because placing it between the legs cannot ensure that it will not fall. Similarly, the egg can be safely retained between the legs with experience, even if the widow engaged in culturally unacceptable behaviour before the rites. Selly also stated:

> "I was told to wash my body to rid his spirit. If I do not, the spirit will stay with me forever. I accepted because I did not want to stay with the spirit forever. It is scary".

Widows had to comply with specific conduct after the rituals were completed. The intention behind the rituals was to inspire fear. In some cases, the widows were told they could not take a different route or shortcut home. Tebby stated:

> "I was told that I cannot use the same way I used to come back, and I was not supposed to use shortcuts".

These are some of the myths that widows are told; in most situations, widows are not told the real reasons for the expected activities. This causes considerable anxiety because there is so much unpredictability around these rites.

### 3.2. Exposure to Harmful Conditions

It was shown that widows were subjected to negative behaviours that can be detrimental to their well-being. The following illustrate some of their experiences:

"...after traditional rituals, I was sitting on the floor for 10 days". (Tebby)

Even though this is an accepted notion, the widow may find it uncomfortable, especially if they are not used to sitting on the floor. It appears that the widow is being punished by this. Another participant indicated:

"As soon as the elders arrived, I was asked if I had eaten or had anything to drink. But when they were called/informed, they then passed on the message that I should not eat or have anything to drink until their arrival. Upon their arrival, they asked me if I had anything to eat or drink. I then said no. They came with their things, and that is when they shot pumpkin seeds in my ears". (Sonto)

Adding to the same point, Juliana highlighted:

"I steamed. I also go incisions with a razor blade when they would take the ash from where you were steaming, and they would smear that on the incision. They also use razors to remove the blood of my husband yes, from my system".

According to the aforementioned extracts, widows are subjected to harsh treatment such as food and liquid deprivation while they wait for rites, as well as the insertion of foreign objects into their bodies and body mutilation in the name of tradition.

Mary and Tebby alleged that they were forced to take cold water baths for an entire year early in the morning. The rationale was that the widow ought to be found cold when the deceased husband shows up to exercise his marital rights.

"I was forced to bathe with cold water every day early in the morning before people started moving around". (Mary)

"It was difficult. I was told that I should bath with cold water for a whole year. Imagine, in winter it was very cold, and I ended up having flu. I had to do it although I knew that it was not good for my health". (Tebby)

Using cold water for a year implies that the widows were forced to endure chilly baths in winter, without thought for their comfort. Zanele mentioned: "The steaming was carried out first to clear the darkness from me". This procedure could have medical repercussions because the widow's physical health may be negatively impacted by some of the herbs used in the steaming. The widows claimed that the goal was to rid themselves of their aura, while others insisted that they needed to get rid of their blood. The removal of the spouse's blood, according to Juliana, is intended to help the widow forget about her deceased husband. Mye demonstrated how eager the widows were to resume their usual lives, which included cooking and eating at home and using their regular utensils, even though they were required to undertake the rituals.

"I could not wait to cook my meals, using my utensils. I was tired of using a plate, cup, spoon which was used by me only. I wanted to be myself again". (Mye)

The desire for a normal life shows that going through widowhood ceremonies is not a pleasant experience and is only carried out to uphold tradition.

### 3.3. Dehumanising Experiences

The widows stated that they felt humiliated and degraded by the widowhood rites. The following remark further demonstrates how dehumanising the rituals were:

"Dithotse is from a pumpkin. They are pumpkin seeds that they crush and then blow into your ear. Once that is done, they then put a doek (Head scarf) on you...The main reason why the elders did what they did is because when someone passes away there is a lot of talk which is not all good. And they did that to protect you, so you do not hear negative things about your husband". (Mbali)

The widow's experience of having her ears dusted with pumpkin powder and then covered with a cloth so she could not hear what people were saying is dehumanising because it makes her feel like an object whose rights can be restricted. This suggests that the widow's close acquaintances may say something she should not hear. Another participant highlighted:

"They said you need to remain home, do not make the mistake of going out into peoples' homes otherwise you are going to give people makghome (this is an illness which is believed to be spread by widows)". (Blessing)

Thato felt that she was not treated well when she went to church. She indicated:

"I was advised by my mother-in-law that while at church and they are worshiping and praising, I should not raise my hands. I had no option except to listen to my mother-in-law who taught me cultural practices and associated traditions".

On the same note, Zanele indicated: "I was supposed to sit at the back in church".

This means that widows are seen as persons who should be isolated when visiting and interacting with others and is linked to contamination. The widow is typically helped to adjust to widowhood through the widowhood rites. However, the widows reported that rather than being helpful, these ceremonies made them feel humiliated and degraded. After the rites, Wish and Preety claimed that they felt denigrated and alone. Wish stated:

"Imagine, having my hair cut, moving around with gowns which were sewn without proper fittings. It was too much for me. I don't know how my children felt when they saw me like this. Do they even think of our children in all this?"

Similarly, Pretty stated:

"One day I went to a taxi, I had to pay my money, and no one wanted to touch my money. If I was not wearing that dress, no one would know that I was a widow. It was terrible".

According to them, after these rituals are over, women usually go back to their regular lives and have to deal with the negative aftereffects of the rites. Selly said:

"I went through hell, and I felt like I was put in a cage. I don't want anything like this to happen to anyone," Selly said, expressing the depth of her anguish.

According to the aforementioned comments, the widow's experiences were so terrible that she felt suffocated and did not want to think of another person going through the same thing.

*3.4. Stigmatisation*

The results show that the widows encountered stigma and discrimination on different occasions.

"My food was dished out and given to me by someone who had already lost her husband. I had my plate/bowl for food and my cup which was placed next to me for tea or water". (Preety)

Adding to the same point, another participant gave her account of having to use a chair specifically meant for her.

"I was not supposed to just sit on any chair, I was supposed to only sit on one specific chair until we had performed an end-of-mourning ceremony". (Tebby)

The widow was fed by a person who is already a widow, which demonstrates how widowhood is perceived as something that may be passed on if one associates with individuals who are not widows. Using kitchenware and furniture that other family members cannot use denigrates widowhood.

"As Africans, there is this misconception that as long as a man is dead, he was killed by the wife, even the family would say that you have killed him, even

when he was Lazarus. But you remain with the pain and only you . . ., when the pain passes them, you still have the pain of their treatment". (Juliana)

The widow in the aforementioned extract was aware that widows are frequently viewed as husband killers. She recognised that in addition to the grief of losing her husband, the way others treat widows makes it even worse. Wish indicated how her in-laws abandoned her after the funeral of her husband.

"But since my husband has passed on, it's been 5 years now, his family had come during the funeral, but since then I have not seen them, you only hear about their ceremonies from the other people ever since my husband died, they don't tell me" (Wish)

The widow's words reveal that she wanted to remain a part of the husband's family after his passing, but the in-laws ignored her whenever they had gatherings.

*3.5. Movement and Social Restrictions*

It was found that the widowhood rites amplify injustice.

"When my husband passed away, I was fully clothed in black. I was not allowed to be out at night. If you had somewhere to go, you had to be taken there and brought back before the sun set. . . They said that those are clothes worn by those who are in mourning". (Mary)

The widow was unable to move freely because of this rite. The fact that she had to be accompanied everywhere she went compromised her privacy as well. There are specific hours when widows are expected to be seated at home. Because of the restriction on their freedom of movement, they are compelled to remain in their homes.

"When my husband passed on, I abstained according to the laws. At 12 'o'clock, I would sit on the floor. And when the clock strikes 12, I work sitting down. That is part of the laws of abstinence. When it hits noon, you are not permitted to be moving around. You must be seated on the floor at that time". (Preety)

Similarly, Juliana believed that going to bed at 11.30 was preferable to delaying sitting down until 12. This incident demonstrates how a widow's freedom might be restricted and how she may be made to do rituals with which she may not be comfortable.

The research also indicated that the mourning process disregards employment. Widows must be aware of their state of sorrow even while at work. Those who commuted a long way from home had to arrive at home before dusk.

"I informed my employer that during my lunch hour, I need to go to my bedroom and pray and sit a bit in the bedroom without going anywhere. I also informed my employer that when I knock off, I need to leave a bit early, so that I am home by 5 pm or 6 pm because of the law of the position I was in at that time". (Selly)

Adding to the same point, another participant had these sentiments:

"I went back to work fully dressed in black. When I got to work, I wore the work overall over the black clothes. At 12 'o'clock, I made sure that 12 did not strike while I was working at my employer's house. I had to go to my bedroom for half an hour or so, just so 12 can pass". (Mbali)

The aforementioned two statements demonstrate how onerous the widowhood ceremonies were for working widows. They interfered with their work and what is expected in the workplace. The rites required the employers to make accommodations for the widows.

**4. Discussion**

*4.1. Instilling Fear and Coercion*

The compulsion to participate in widowhood rites because of fear suggests that the majority of the participants were obedient to their culture since they were unable to challenge or defy it. The intersectionality view shows how the cultural view prevents an independent analysis of oppression. The widows feared social exclusion more than

anything else. The results demonstrated that extreme fear is used to terrify widows. The widows expressed their lack of say in these widowhood rites. They were not allowed to decide whether to carry out the rites. Widows are advised to stay away from crowded areas while they are in mourning. The widow becomes isolated because of these expectations. The in-laws' role in widowhood customs cannot be disregarded because they frequently have a say in them. Dube (2022) argues that in-laws are opportunistic as they use cultural norms to prolong violence against the bereaved widows since isolation is often imposed on the widows by their in-laws. Gender dynamics are important in developing coercive tactics and establishing fear because widows are women. The intersection of religious and cultural values shapes coercive practices and creates anxiety among widows. Some traditions or rituals may be intended to evoke fear and compliance. Although the findings support Dube's (2022) claims, the in-laws' actions cannot be attributed solely to the widowhood ceremonies. They are the guardians of a custom that has been practiced for many years. Some, however, tend to go overboard. The dominance and oppressive actions of the in-laws impact the widows as they experience multiple marginalisation, that is, being a woman, a widow, and a widow with no support system. If these rites are performed with malicious intent or to settle scores, they are violating the widow's rights. The widows would be even more afraid because of the lack of transparency surrounding the justifications for these actions.

*4.2. Exposure to Harmful Conditions*

It is clear from the extracts that the widows underwent rites that were detrimental to their health and well-being for the sake of purification. According to a study by Ajayi et al. (2019), some widowhood rites are harmful to widows on all fronts, including psychologically, physically, socially, and emotionally. Similarly, Dube (2022) concluded that the many forms of isolation that widows suffer have a detrimental impact on the well-being of the widows and their children on a variety of psychological, social, physical, and economic levels. Although the findings are consistent with those of Dube (2022) and Ajayi et al. (2019), it may be argued that although the widows in the study emphasised the risks to their health, there was never a case where one was physically hurt due to these ceremonies. However, the psychological and emotional repercussions were evident in the interviews when the women were telling their stories. The widows' ability to eloquently describe their stories suggests that they are still fresh in their thoughts. The type and intensity of harmful conditions widows experience may vary depending on age-related expectations and preconceptions. Younger widows with hopes of becoming married again will probably be more affected by these purification rituals because part of them is intended to purge the widow of the man's spirit.

*4.3. Dehumanising Experiences*

Traditional treatment of widows demonstrates how dehumanised they feel. According to Dube (2022), after the rites, widows experience a loss of self-worth, dignity, and self-esteem and find it challenging to interact with other community members. From an intersectional perspective, the widows are the ones who understand their oppression the most because of their lived experiences; as a result, they draw attention to the fact that they feel humiliated and less human. Due to expectations and gender prejudices, widows may experience dehumanization. These degrading customs during widowhood rites are also influenced by religious and cultural norms. In churches, where widows may be forced to quit engaging in activities, the dehumanisation of widows was on display. The participants claimed that some widows were advised not to raise their hands during times of praise and worship; this subtly reinforced the position of widows as second-class people. However, one widow indicated that her faith helped her to go through mourning. It may be claimed that the church can helpfully aid in the grieving process. The fact that widows attend church during their time of grieving demonstrates their faith in the support they receive there. Even if widows receive consolation in church, this must be given respectfully so that

the widows do not feel in any way degraded. The widows' psychological and emotional well-being is impacted if they are required to sit in particular places in church or are excused from participating in specific activities.

### 4.4. Stigmatisation

Widows were required to continue abstaining after being cleansed, and this demonstrates that the widow is denied human rights despite the cleansing. Contrary to what is intended, widows are further denied their right to association, even though the cleansing's primary goal is to enable widows to lead normal lives. They still encounter difficulties because others avoid them because they are said to contain darkness that they can spread to others. The stigma that widows experienced was influenced by the intersection of the widow's age, gender, and cultural and religious identity. For example, discriminatory actions are reinforced by the intersection of patriarchal standards, cultural views, and religious beliefs. Munala et al. (2022) reiterates that a widow is perceived in the community as possessing a bad omen because of the death of her husband. Many who were close to the widows were hesitant to lend them objects such as glasses, plates, or even chairs, which demonstrates the stigma society places on widows. In-laws would stop communicating with the widows since they were accused of killing their spouses. This further stigmatises widowhood because becoming a widow is linked to a husband's murder. Being associated with such a suspicion ensures that the widow will always bear the burden of responsibility for her husband's passing, which is what society typically identifies with widowhood. The way widows are treated has an impact on their well-being because they are frequently victims of violence of some kind.

### 4.5. Movement and Social Interaction Restrictions

Widows experienced various emotional and social challenges when their movement and social interactions were restricted. This meant that they felt isolated, exacerbating the already existing sense of loss and grief (Ratcliffe 2022). If they are restricted in their movement and social interaction, they cannot engage in activities that promote emotional well-being such as exercise, socialising, or pursuing hobbies. This, furthermore, denies them the emotional support of friends and family. It is, however, important to note that these effects are highly individualised as different individuals may respond differently to similar circumstances based on their intersectional identities.

Due to cultural conventions that may dictate particular behaviors and constraints for widows, women are frequently disproportionately impacted by social and mobility restrictions. The norms unique to the community that influence the limitations placed on widows may have a major impact on this. Some societies may enforce stricter regulations due to their cultural and religious beliefs.

The constant persecution of widows means that the widows lack somebody to confide in since the shame associated with widowhood arouses memories of the burial. The stigma associated with widowhood raises the possibility of losing acquaintances and relationships with significant residents of the community (The London School of Economic and Political Sciences 2021). The widows become even more isolated as a result and being alone can make widows depressed. It is necessary to address the conflicts that arise between traditional cultural beliefs and the proper way to respond to women's rights. The patriarchal impact was clear, as the widow is expected to carry out the rites to be released from the dead husband's dominance and to be subservient to him even in death. This demonstrates unequivocally that the wife's spouse has a significant impact on her life after his death. This falls short of Sustainable Development Goal 5, whose main focus is the abolition of all types of violence against women as well as detrimental customs such as widowhood ceremonies, among other things (Ajayi et al. 2019). Most participants expressed considerable trepidation about the widowhood rites, which they felt infringed on their human rights. Although widowhood is a homogenous group, different widows may

experience widowhood differently. The different rites applied to the widows exacerbate inequality among women and thus produce an oppressive system.

Despite the existence of multiple human rights bodies at international and national levels, the custodians of traditional culture may take pleasure in and safeguard oppression (Ajayi et al. 2019). Widows can only advocate for their rights if they are not bound by the prescripts of their culture. It is important to note that not all widows experienced the harsh components of these rites because some still adhere to their cultures and would prefer to follow customs. Having widows narrate their experiences during their mourning period provided insight into the attitude of the community towards widows. They confirmed the negative attitudes of pity and fear shown towards them, as well as the idea of darkness. In their state of mourning, considering that it is common for widows to wear black to show that they are mourning their deceased husbands, many widows spoke about the negative attitudes shown by the community. Because they view widowhood activities as cultural rituals intended to pay their final respect to their deceased spouses, victims do not disclose such cultural abuses officially or formally (Ajayi et al. 2019). One would question why widows endure these harmful widowhood rites. The women decided to submit themselves to the rites despite a seemingly widespread awareness that these procedures are detrimental and a violation of their rights out of respect for tradition, safety from social backlash, and concern for the security of their offspring (Ajayi et al. 2019). As a result, widowhood rites are discriminatory, demeaning, and disparaging because they diminish the widow's value.

### 5. Conclusions

The article looked at how some widows in South Africa are subjected to violent widowhood rites.

- It was found that widows were forced to participate in widowhood rites through pressure and fear.
- The widows' health is negatively impacted by these rites, which also run the risk of stigmatising them.
- The human rights of these widows are not upheld. Therefore, it is recommended that laws be promulgated to defend widows' rights and shield them from predatory cultural ideals.

There is therefore a need to enforce the recommendations of SDG 5 so that widows are not discriminated against and exposed to harmful practices. The empowerment of widows is necessary so that they can speak out against any customs with which they may not agree. Further investigation is required to determine why some widows continue to consent to these widowhood rites despite the findings showing that the majority are not comfortable with them.

**Author Contributions:** Conceptualisation, R.S.; Methodology, R.S.; Investigation, R.S. and S.D.; Data Curation, R.S. and S.D.; Writing-original draft preparation, R.S.; Supervision, S.D. All authors have read and agreed to the published version of the manuscript.

**Funding:** This research was funded by Women in Research grant and the APC was paid by the University of South Africa.

**Institutional Review Board Statement:** The study was conducted in accordance with the Declaration of Helsinki and approved by the University of South Africa College of Education Ethics Review Committee (2018/06/13/90167260/44/MC).

**Informed Consent Statement:** Written informed consent was obtained from all participants involved in the study.

**Data Availability Statement:** Data is unavailable due to privacy or ethical restrictions.

**Conflicts of Interest:** The authors declare no conflict of interest.

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
