# Peer review of "The Violent Aspect of Widowhood Rites in the South African Context"

_socsci, doi:10.3390/socsci13020115_

Round 1

Reviewer 1 Report

Comments and Suggestions for Authors

The interesting topic of the violent treatment of widows in South Africa was raised. The text emphasizes this thesis - about violence used - in a selected cultural circle - against women taking on the role of widows. The adoption of the intersectionality perspective was declared - as useful for explaining discrimination against women as a result of the "intersection" of the woman's identity, and the identity of a widow, and the identity of a community member (the Gauteng province of South Africa), in which the individual was socialized. The question arises about the privileges that women may experience as a result of such a combination of factors (gender, widowhood, community, and human status/identity in the contemporary world). (I mean the Sustainable Development Goal 5 (SDG 5) referred to in the text, which seeks to eliminate all forms of violence and harmful behaviors against women and girls (UN 2015).) The experiences of the surveyed women referred to in the text may constitute the basis for inferring evolution in this area: the emergence of opportunities for the discussed group - as a result of the intersection of many identities - to obtain the privilege of obtaining the status of an autonomous person (understood as an achieved identity).

One may wonder whether it would be more accurate to refer to the idea of Multiple Discrimination.

The choice of theoretical framework seems appropriate, but I feel there is a certain deficiency (or ambiguity) in explaining the obtained research results by referring to this framework.

Perhaps it would be worth clearly defining the categories of manifestations of discrimination experienced by widows in the theoretical foundations (first part of the article): 1. Instilling fear and cohesion; 2 - Exposure to harmful conditions; 3- Dehumanizing experiences, 4 – Stigmatization; 5- feeling injustice.

This may also be important for improving the clarity of the methodology description. The authors provided little information about the questions asked to the respondents. The method of selecting response categories or including respondents' statements into categories was not provided. Referring to qualitative research does not necessarily exclude providing at least quantitative/numerical/percentage summaries of the responses obtained. No data were provided to describe the study group - apart from the information that they were widowed women.

Due to the accentuated cultural diversity of the region in which the research was conducted, descriptions of discrimination are diverse, which can be interpreted as an effect of random selection and raises questions about the ethnic structure, diverse paths, and socialization patterns. Enrichment with basic quantitative analyses could make it easier for readers to understand the arguments presented in the article.

Perhaps it would be good to consider modifying, for example, the ending, which is clearly influenced by one of the cited articles ( Dube, Misheck. 2022. Isolation and its impact on widows: Insights from low-resourced communities in Binga district, 590 Zimbabwe. Social Sciences, 11:298. https://doi.org/10.3390/socsci11070298).

Formulating the main conclusions in bullet points would improve the quality of the presentation.

The article raises important issues and calls for socio-cultural change in the southern African region. In my opinion, it requires refinement in the above-mentioned issues.

Lack of information on compliance with the rules of ethics in the discipline

Author Response

Please find attached the reviewed article, and the corrections.

Reviewer 2 Report

Comments and Suggestions for Authors

The authors need to improve the objective of the study, the methodology, namely what kind of procedure (ex. MAXQDA or other) used to analyse the results and organized better the results. 

Reviewer 3 Report

Comments and Suggestions for Authors

The abstract needs rewriting to focus on the research question that was examined, the findings and its contribution to the existing scholarship.  

The section 1.2 on widowhood rites, and section 1.3 on violent forms of widowhood rites present a somewhat descriptive, and occasionally repetitive, account of the widowhood rites. 

Ethical issues need to be addressed – for example, what measures were undertaken to manage the risks associated with participation in the research, and the impact of recalling traumatic events, particularly if the participants are still living in an abusive context? Was ethical approval sought. 

Sample, inclusion criteria was widowhood, but as focus specifically on the period/incidents of widowhood rites rather than a longer term focus on life as a widow, it would be useful to indicate when the research participants became a widow, to enable reflection on any changes or continuities in their traditions over time. 

The authors use the term ‘discomfort’ (eg., in the abstract) to signal the coercion entailed in these rituals – a clearer engagement with the concepts and language commonly utilised in the scholarship on violence against women will strengthen this work. The coercive nature of these rituals (the social contexts and constraints within which consent is constructed), the harm entailed in the rituals and the impact on those undergoing them could be more clearly conceptualised. The abstract refers to the persecution of those who resist these rituals – but there is no discussion of this in the main article. 

The abstract refers to intersectionality, but there is little engagement within the article to how gender intersects with other social relations of power to shape the nature and impact of the widowhood rituals. The early section on intersectionality (1.4) presents a simplistic and sometimes not entirely clear account of the key tenets of intersectionality – it would be good to see an outline of the way in which the concept of intersectionality is deployed by the authors in their research and to see evidence of an intersectional lens in the presentation (for example, there is no indication of the social location of the research participants – age, class position, etc which might enable a differentiated analysis of their experiences of the research findings and its analysis in the discussion. At the moment there is little evidence of how gender intersects with other social relations of power in shaping the nature and impact of widowhood rites, or indeed any resistance to such rites. 

Typos: section 1.3 subtitle – should be coercion not cohesion?  

On the whole, the article is overwhelmingly descriptive, with the discussion section reiterating and outlining the findings, rather than analysing their implications for how we conceptualise VAW or for policy/practice responses to these rituals.

Comments on the Quality of English Language

The quality of English is fairly competent, but the writing is somewhat simplistic.

Reviewer 4 Report

Comments and Suggestions for Authors

Comments on the Quality of English Language

Editing for English language is needed. 

Round 2

Reviewer 1 Report

Comments and Suggestions for Authors

After refinement, the text is more interesting and contains an in-depth analysis of the socio-cultural situation in the explored area of Africa.

Extending the part presenting the research results and introducing quotations from the collected material allows for a better understanding of the idea of the article.

An issue arises regarding the content of the question or introduction to the conversation with the respondents. Were you asked, for example: Tell me about your experiences in the first days after your husband's death? Were any questions asked?

Reflection on widows' submission to cultural "rites of passage": 

The very fact of the husband's death (entering the social role of a widow) is not comfortable due to the loss of a husband-guardian, intimate partner (?), father of children, loss of the status of a wife, and a radical change in the economic and legal situation. Perhaps women simply experience these rites of passage as vexing, unpleasant, but necessary to achieve their new changed social status as widows.

I suggest checking the way the bibliography is written.

Reviewer 4 Report

Comments and Suggestions for Authors

This revised manuscript is much improved. My concerns were addressed to my satisfaction. 

Comments on the Quality of English Language

I did see a few spelling errors that need to be corrected (e.g., flue rather than flu and bath rather than bathe in a quote). 
